# Optimal Light Dose for hEGFR-Targeted Near-Infrared Photoimmunotherapy

**DOI:** 10.3390/cancers14164042

**Published:** 2022-08-22

**Authors:** Hideyuki Furumoto, Ryuhei Okada, Takuya Kato, Hiroaki Wakiyama, Fuyuki Inagaki, Hiroshi Fukushima, Shuhei Okuyama, Aki Furusawa, Peter L. Choyke, Hisataka Kobayashi

**Affiliations:** Molecular Imaging Branch, Center for Cancer Research, National Cancer Institute, National Institutes of Health, Bethesda, MD 20892, USA

**Keywords:** near-infrared photoimmunotherapy, light dose, EGFR, side effects

## Abstract

**Simple Summary:**

Near-infrared photoimmunotherapy (NIR-PIT) is a cancer therapy that selectively destroys target cells by first injecting monoclonal antibodies conjugated with a photon absorber (IRDye700DX) into the subject and then activating it at the tumor site by applying nonthermal doses of NIR light at 690 nm. NIR-PIT causes immediate immunogenic cell death but also induces a slightly delayed activation of anti-tumor host immunity which can result in complete responses. The immediate therapeutic effect of NIR-PIT can be enhanced by increasing the dose of near-infrared light irradiation; however, this can cause local side effects such as edema. Since the activation of host immunity also adds to the anti-tumor effect it might be possible to reduce the light dose to avoid immediate side effects while maintaining efficacy of the therapy. In this study, we varied the light dose needed to achieve the maximum therapeutic effect in an immunocompetent mouse model. We show that higher-than-needed light doses caused significant local transient edema that could be avoided with lower but still effective light doses. Here, we present our strategy for optimizing the light dose for NIR-PIT.

**Abstract:**

Near-infrared photoimmunotherapy (NIR-PIT) is a newly developed cancer therapy that targets cancer cells using a monoclonal antibody-photon absorber conjugate (APC) that is bound to the target cell surface. Subsequent application of low levels of NIR light results in immediate cancer cell death. The anti-tumor effect of NIR-PIT in immunocompromised mice depends on immediate cancer cell death; therefore, the efficacy increases in a light-dose-dependent manner. However, NIR-PIT also induces a strong anti-tumor immune activation in immunocompetent mice that begins soon after therapy. Thus, it may be possible to reduce the light dose, which might otherwise cause local edema while maintaining therapeutic efficacy. In this study, we determined the optimal dose of NIR light in NIR-PIT based on a comparison of the therapeutic and adverse effects. Either one of two monoclonal antibodies (mAbs) against human epidermal growth factor receptor (hEGFR), Cetuximab or Panitumumab, were conjugated with a photo-absorbing chemical, IRDye700DX (IR700), and then injected in hEGFR-expressing mEERL (mEERL-hEGFR) tumor-bearing C57BL/6 immunocompetent mice or A431-GFP-luc tumor-bearing athymic immunocompromised mice. NIR light was varied between 0 to 100 J/cm^2^ one day after administration of APC. In an immunocompromised mouse model, tumor growth was inhibited in a light-dose-dependent manner, yet extensive local edema and weight loss were observed at 100 J/cm^2^. On the other hand, in an immunocompetent mouse model using the mEERL-hEGFR cell line, maximal tumor response was achieved at 50 J/cm^2^, with a commensurate decrease in local edema. In this study, we show that a relatively low dose of NIR light is sufficient in an immunocompetent mouse model and avoids side effects seen with higher light doses required in immunocompetent mice. Thus, light dosing can be optimized in NIR-PIT based on the expected immune response.

## 1. Introduction

Near-infrared photoimmunotherapy (NIR-PIT) is a newly developed cancer therapy that selectively and locally destroys cancer cells. NIR-PIT is a targeted therapy in which a water-soluble silica phthalocyanine dye, IRDye700DX (IR700), is conjugated to an antigen-specific monoclonal antibody (mAb) to form an antibody-photon absorber conjugate (APC) that binds with high affinity to target molecules on the cell membrane [1]. The APC is then activated at the tumor by the application of NIR light (690 nm) which causes cancer cell death. Upon exposure to NIR light, IR700 undergoes a profound photochemical change, releasing ligands that convert the molecule from highly hydrophilic to highly hydrophobic. Such rapid changes cause antibody aggregation and damage to the cell membrane, resulting in cell swelling and rupture [2,3]. Unlike conventional therapies that usually induce apoptosis, NIR-PIT induces a rapid immunogenic cell death, resulting in the stimulation of the host’s immune response against the tumor [4]. Therefore, NIR-PIT damages cancer cells using two mechanisms: direct killing of cells and indirect killing due to the enhancement of the host’s immunity.

NIR-PIT has been successfully used in human clinical trials. A phase III clinical trial of NIR-PIT targeting human epidermal growth factor (hEGFR) is currently underway in patients with inoperable head and neck squamous cell carcinoma (HNSCC) (https://clinicaltrials.gov/ct2/show/NCT03769506, accessed on 1 June 2022). Cetuximab-IR700 (ASP1929) and a NIR laser device (BioBlade, Rakuten Medical, Inc., San Diego, CA, USA) were conditionally approved by the Pharmaceuticals and Medical Devices Agency (PMDA) in Japan, in 2020, for the treatment of unresectable HNSCC. Thus, light dosing is of importance in mitigating the potential side effects of NIR-PIT.

Several parameters can affect the balance between efficacy and adverse events [5,6,7,8,9]. NIR light intensity is an important factor in determining the effectiveness of NIR-PIT, depending on the conditions such as the type of light source, power density, and total dose [10,11,12,13]. Increased cytotoxicity in vitro and in vivo have achieved with increasing light doses. For in vivo animal experiments, NIR-PIT studies have typically been performed at 30–100 J/cm^2^ of NIR light. Even low NIR light doses are sufficient to exert cytotoxic effects; however, a higher light dose can deliver more light to the deeper parts of the tumor; thus, higher light doses are often selected to maximize the therapeutic efficacy [10]. On the other hand, in clinical settings, some patients experience temporary acute edema around the tumor after treatment, which appears to be related to the light dose. In NIR-PIT, reactive oxygen species (ROS) are produced by the reaction of NIR light and IR700 on unbound APCs under oxygen-rich conditions. This ROS is thought to be the cause of acute edema, and an excessive dose of NIR light could intensify acute edema [14]. Therefore, it is important to perform treatment with a proper light dose that is sufficient and not excessive.

The impact of different light doses in NIR-PIT has been well studied in immunocompromised mice. In an immunocompromised mouse model, the anti-tumor effect solely depends on the direct cytotoxicity, and the efficacy increases in a light-dose-dependent manner. However, in immunocompetent mice, less light is needed, because the anti-tumor effect of NIR-PIT comes not only from the direct cell killing but also from the immune activation following the NIR-PIT. Recently, an hEGFR-expressing murine oropharyngeal HPV-associated cancer cell line (i.e., mEERL-hEGFR) was established. This is an ideal mouse tumor model to simulate the clinical setting of hEGFR-targeted NIR-PIT, because we can use the same mAbs as in clinical settings. Additionally, since hEGFR is not expressed in mouse host cells, immune responses after NIR-PIT can be assessed [15].

In this study, we determined the optimal NIR light dose in hEGFR-targeted NIR-PIT in immunocompetent mice, with the aim of maximizing the therapeutic effects and minimizing the adverse events.

## 2. Materials and Methods

### 2.1. Reagent

Water-soluble, silica-phthalocyanine derivative, IRDye700DX NHS ester (C_74_H_96_N_12_Na_4_O_27_S_6_Si_3_; molecular weight of 1954.22), was obtained from LI-COR Bioscience (Lincoln, NE, USA). Cetuximab, a chimeric (mouse/human) monoclonal antibody (mAb) directed against hEGFR, was purchased from Bristol-Meyers Squibb Co. (Princeton, NJ, USA). Panitumumab, a fully humanized IgG_2_ mAb directed against hEGFR, was purchased from Amgen (Thousand Oaks, CA, USA). All other chemicals were of reagent grade.

### 2.2. Synthesis of IR700-Conjugated Cetuximab and Panitumumab

Cetuximab (1 mg, 6.6 nmol) or Panitumumab (1 mg, 6.8 nmol) was incubated with IR700 NHS ester (66.8 μg, 34.2 nmol, 5 mmol/L in DMSO) in 0.1 mol/L Na_2_HPO_4_ (pH 8.5) at room temperature for 1 h. The mixture was purified with a Sephadex G50 column (PD-10; GE Healthcare, Piscataway, NJ, USA). The protein concentration was determined with the Coomassie Plus protein assay kit (Thermo Fisher Scientific Inc., Rockford, IL, USA) by measuring the absorption at 595 nm with spectroscopy (8453 Value System; Agilent Technologies, Santa Clara, CA, USA). A concentration of IR700 was measured by absorption at 689 nm with spectroscopy to confirm the number of fluorophore molecules conjugated to each mAb. The synthesis was controlled so that an average of three IR700 molecules were bound to a single antibody. The IR700-conjugated Cetuximab and panitumumab are abbreviated as Cet-IR700 and Pai-IR700, respectively.

### 2.3. Cell Lines and Culture

Parental mEERL cells were established by transduction of HPV 16 E6/E7 and hRAS to C57BL/6-derived oropharyngeal epithelial cells [16,17,18]. The mEERL-hEGFR cells were cultured in DMEM/F-12 (Thermo Fisher Scientific) supplemented with 10% fetal bovine serum (FBS), 100 IU/mL penicillin–streptomycin (Thermo Fisher Scientific), and 1× human keratinocyte growth supplement (Thermo Fisher Scientific), which was modified from a previous report [15,19]. In addition, MDAMB468-GFP-luc and A431-GFP-luc cells that highly express EGFR and stably expressed GFP and luciferase were grown in RPMI 1640 supplemented with 10% FBS and 100 IU/mL penicillin–streptomycin in tissue culture flasks in a humidified incubator at 37 °C in an atmosphere of 95% air and 5% carbon dioxide.

### 2.4. Animal and Tumor Models

All in vivo procedures were conducted in compliance with the Guide for the Care and Use of Laboratory Animal Resources (1996), US National Research Council, and approved by the local Animal Care and Use Committee (MIP-003; project number: P183735). Six- to eight-week-old female C57BL/6 mice were purchased from The Jackson Laboratory (Bar Harbor, ME, USA). During procedures, mice were anesthetized with isoflurane. One million mEERL-hEGFR cells were inoculated into the right side of the dorsum of the mice. Six- to eight-week-old female homozygote athymic nude mice were purchased from Charles River (Frederick, MD, USA). Two million A431 cells were inoculated in the same manner as the C57BL/6 mice. The hair overlying the tumor site was removed before light exposure and the imaging studies. Tumor volume was estimated using the ellipsoid formula: (major axis) × (minor axis)^2^ × 0.5. Tumor volumes were measured twice a week until the volume reached 1000 mm^3^, whereupon the mice were euthanized with CO_2_.

### 2.5. In Vitro hEGFR Expression Analysis

To evaluate hEGFR expression on mEERL-hEGFR, A431-GFP-luc, and MDAMB468-GFP-luc cells in vitro, 0.2 × 10^6^ cells were stained with PE-labeled anti-hEGFR Ab (clone AY13, BioLegend, San Diego, CA, USE) or its PE-labeled mouse IgG1κ isotype control (clone MOPC-21, BioLegend). They were incubated with the fixable survival dye (Thermo Fisher Scientific) at 4 °C for 30 min. Cell fluorescence was then analyzed by BD FACSLyric (BD Biosciences, Franklin Lakes, NJ, USA) and FlowJo software (BD Biosciences).

### 2.6. In Vitro NIR-PIT

For quantitative assessment of cytotoxicity, two hundred thousand cells of mEERL-hEGFR, A431-GFP-luc, or MDAMB468-GFP-luc were seeded into 12-well plates. After one day, the cells were incubated with 10 μg/mL of Cet-IR700 for 1 h at 37 °C. After washing with PBS, phenol-red-free medium was added. The NIR light (690 nm, 150 mW/cm^2^) for the mEERL-hEGFR cells was applied at 0, 30, 50, or 100 J/cm^2^, while NIR light doses for A431-GFP-luc and MDAMB468-GFP-luc cells were 0, 10, 25, 50, or 100 J/cm^2^. One hour after light exposure, the cells were collected with trypsin and stained with 1 µg/mL propidium iodide (PI). The percentage of PI-stained cells were analyzed with flow cytometry (FACSCalibur, BD Biosciencse) and FlowJo software (BD Biosciences).

### 2.7. In Vivo NIR-PIT

The mEERL-hEGFR and A431-GFP-luc tumor-bearing mice were randomly divided into 4 and 5 groups, respectively, with 10 animals each for the following treatment: mEERL-hEGFR: (1–4) 100 μg of Cet-IR700 with 0, 25, 50, or 100 J/cm^2^ NIR light; A431-GFP-luc: (1–5) 100 μg of Pan-IR700 with 0, 10, 25, 50, or 100 J/cm^2^ NIR light irradiation, respectively. Cet-IR700 or Pan-IR700 was intravenously injected through the tail vein. Twenty-four hours after administration, NIR light (i.e., 690 nm, 150 mW/cm^2^) was applied to all tumors. The laser device used was a ML7710 system (Modulight Corporation, Tampere, Finland), and the NIR light was changed to a parallel beam via a collimator. The surface of the mouse other than the tumor was covered with aluminum foil, and holes approximately 0.5 inch in diameter were drilled to avoid exposing normal tissue to NIR light. Acute cell killing was evaluated with bioluminescence imaging (BLI). 

### 2.8. IR700 Fluorescence Imaging Study

IR700 fluorescence images were obtained with a Pearl Imager (LI-COR Bioscience, Lincoln, NE, USA). Images were obtained with the 700 nm fluorescence channel, and ROIs were manually drawn on both the right dorsum (i.e., tumor) and the left dorsum (i.e., background). The average fluorescence intensity of each Regions of interest (ROI) was measured using Pearl Cam Software (LI-COR Bioscience), and the target-to-background ratios (TBRs) of the fluorescence intensities were calculated according to previous reports.

### 2.9. Bioluminescence Imaging (BLI) Study

For BLI, 200 μL of 15 mg/mL D-luciferin (Gold Biotechnology, St. Louis, MO, USA) was injected intraperitoneally, and imaging was performed 5 min later (PhotonIMAGER; Biospace Lab). Light intensity per unit area and time were quantified by placing ROIs on the tumor implanted on the right dorsum and a background ROI over the corresponding left dorsum. TBR was calculated before and after NIR-PIT. To estimate the effective area of the NIR-PIT, the light intensity was measured on a pixel-by-pixel basis and then compared to the background.

### 2.10. Magnetic Resonance Imaging (MRI) Study

MRI was used to show the degree of edema in the treated tumor region. NIR-PIT was performed according to the in vivo procedure described above, and 24 h later, the mice were anesthetized with pentobarbital. MRI was performed on a 3-T scanner using an in-house 10-inch-circle-shaped mouse receiver coil array (Elition 3T; Philips Medical Systems, Best, The Netherlands). Scout images were obtained to accurately locate the tumor. All mice underwent T2WI fat-sat. All images were obtained in the coronal plane. All images were analyzed using Image J software version 1.53r (Bethesda, MD, http://rsb.info.nih.gov/ij/, accessed on 1 May 2022). The high signal intensity area derived from each image in T2WI fat-sat and short TI inversion recovery (STIR) was calculated using Image J.

### 2.11. Statistical Analysis

Data are expressed as the mean ± standard error. Statistical analyses were carried out using GraphPad Prism version 7 (GraphPad Prism; GraphPad Software Inc., La Jolla, CA, USA). For multiple comparisons, one-way analysis of variance (ANOVA) followed by Tukey’s test was used. The Student’s *t*-test was also used for comparison between two groups. *p*-values less than 0.05 were considered statistically significant.

## 3. Results

### 3.1. In Vitro NIR-PIT with Cetuximab-IR700 (Cet-IR700)

mEERL-hEGFR, A431-GFP-luc, and MDAMB468-GFP-luc showed expression of hEGFR on the cell surface, and A431-GFP-luc and MDAMB468-GFP-luc showed high expression. The expression of mEERL-hEGFR was sufficient but poor compared to the other two cell lines (Figure 1A). The cytotoxic effects of hEGFR-targeted NIR-PIT using Cetuximab among the three cell lines (i.e., mEERL-hEGFR, A431-GFP-luc, and MDAMB468-GFP-luc) were evaluated in vitro. Cytotoxicity after NIR-PIT was measured by counting the number of dead cells stained with PI by flow cytometry. Cet-IR700 alone or NIR light alone did not produce cytotoxicity. The percentage of PI-stained dead cells increased in a NIR light-dose-dependent manner (Figure 1B). In A431-GFP-luc cells, almost all cells were killed at 50 J/cm^2^ and 100 J/cm^2^. This showed that the cytotoxicity depended on the NIR light dose in the various cell lines expressing hEGFR. Moreover, the degree of hEGFR expression was proportional to the cell-killing effect of NIR-PIT in vitro.

### 3.2. In Vivo NIR-PIT with Panitumumab-IR700 (Pan-IR700) in Immunocompromised Mice

To investigate how different light doses affected the therapeutic effect of NIR-PIT in immunocompromised mice, we performed NIR-PIT in A431-GFP-luc xenografts using Pan-IR700, because Pan-IR700 is more effective in immunocompromised mice than Cet-IR700, as shown in a previous study, due to the fact of its long circulation half-life and lack of host immunity [5]. Treatment and imaging schedules are shown in Figure 2A. Fluorescence imaging showed the accumulation of Pan-IR700 in A431-GFP-luc tumors. After NIR light irradiation, IR700 fluorescence in A431-GFP-luc tumors diminished at all light doses. While some IR700 fluorescence remained after irradiation with 25 J/cm^2^ NIR light, fluorescence was completely lost at doses above 50 J/cm^2^ (Figure 2B). TBR decreased in a light-dose-dependent manner but did not decrease further above 50 J/cm^2^ (Figure 2C). The cellular activity of the A431-GFP-luc cells was monitored by BLI (Figure 2D). The BLI signal of the NIR-PIT group exposed to light doses higher than 25 J/cm^2^ decreased the day after treatment and gradually increased after 2 days, indicating tumor regrowth. The BLI signal of the NIR-PIT group irradiated with more than 50 J/cm^2^ NIR light was significantly lower than that of the 0 J/cm^2^ and 10 J/cm^2^ groups 5 days after treatment (Figure 2E).

Tumor growth was inhibited in a light-dose-dependent manner (Figure 3A). Survival was significantly prolonged in the 100 J/cm^2^ group compared with the 0 J/cm^2^ group (Figure 3B).

Next, we evaluated edema formation at the treatment site. One day after NIR-PIT, the edema-affected area enlarged in a light-dose-dependent manner from 25 J/cm^2^. The skin around the treatment site was also paler in color in the 100 J/cm^2^ groups indicating possible decreased blood flow (Figure 4A). Additionally, after treatment, the weight of the mice gradually decreased, indicating a systemic insult to the animal. The largest weight loss was always observed on the seventh day after irradiation, and the amount of weight loss increased in a light-dose-dependent manner (Figure 4B). In the xenograft mouse model, the therapeutic effect and the intensity of the side effects depended on the light dose.

### 3.3. In Vivo NIR-PIT with Cet-IR700 in Immunocompetent Mice

The anti-tumor effects and side effects were then evaluated in immunocompetent mice. In order to simulate a superior clinical situation, the mEERL-hEGFR tumor model was treated with Cet-IR700 followed by NIR light. In this experiment, the anti-tumor effect of NIR-PIT came both from direct cancer cell killing and immune activation. The treatment and diagnostic imaging regimens are shown in Figure 5A. IR700 fluorescence from the tumor on the right dorsal side was monitored for IR700 photo-bleaching after light exposure. TBR was evaluated to quantify the fluorescence intensity of Cet-IR700 before and after irradiation. Fluorescent signals accumulated in the mEERL-hEGFR tumors, indicating that Cet-IR700 was successfully delivered and bound to the mEERL-hEGFR tumors. The IR700 fluorescence signal in the mEERL-hEGFR tumors completely disappeared after exposure to NIR light at or above 50 J/cm^2^ (Figure 5B), and TBR was significantly reduced compared to before the NIR-PIT in the 25 J/cm^2^ group and at higher light doses. However, TBR after NIR-PIT was minimally different between the 50 and 100 J/cm^2^ groups. (Figure 5C).

There was statistically no significant difference in tumor growth among the groups of 25 J/cm^2^ or higher light doses. In the 50 and 100 J/cm^2^ groups, tumor growth was similar but tended to be more suppressed than in the 25 J/cm^2^ group (Figure 6A). The survival rate of mice treated with 25 J/cm^2^ or more NIR light was greater than that of the 0 J/cm^2^ treatment group (Figure 6B). There was no significant difference in survival among the three groups exposed to 25 J/cm^2^ or more, but complete remission was observed in 1 out of 10 mice in both the 50 and 100 J/cm^2^ groups. 

Weight change and edema were evaluated as side effects of treatment. Stress in mice was monitored by loss of body weight. T2WI fat-sat MRI quantified the degree of edema in the mice on the day after irradiation. The area and severity of edema increased in a light-dose-dependent manner, and skin pallor was observed in the 100 J/cm^2^ groups (Figure 7A–C). In the treated mice, body weight increased the day after treatment in the high-light dose groups due to the temporary edema. However, after one week, body weight in the 100 J/cm^2^ treatment group was less than that in other groups (Figure 7D). These side effects intensified in a dose-dependent manner.

## 4. Discussion

In this study, we aimed to determine the optimal light dose, balancing therapeutic efficacy and adverse effects in immunocompromised and immunocompetent mouse models. 

Previous studies using immunocompromised mice have shown that increasing the NIR light dose increases cell killing with NIR-PIT [10,20]. This observation was confirmed in this study in which tumor growth inhibition and long-term survival was enhanced in a light-dose-dependent manner in immunocompromised mouse models. However, in immunocompetent mice, NIR-PIT at low or high exposures of NIR light showed similar anti-tumor effects because of the added cytotoxicity caused by immune activation. In immunocompetent mice, the anti-tumor effect after NIR-PIT comes not only from the direct killing of cancer cells but also from immune-mediated cell killing. We previously reported post-NIR-PIT immune activation in the same model as this study: hEGFR-targeted NIR-PIT against mEERL-hEGFR tumor using Cetuximab as an APC [5]. This demonstrated that the host’s anti-tumor immune response was promoted after NIR-PIT [4,21]. Thus, at light doses at or approximately 50 J/cm^2^ or above, the effect of host immunity has probably exceeded the benefit of increasing cancer cell killing with increased light doses. Indeed, such combinations of direct cell killing, immune-mediated cell killing, and modest light doses led to cures in some of the mice.

IR700 is not only a photon absorber required for NIR-PIT, but its fluorescence may also be utilized to monitor its therapeutic effects. Fluorescent images immediately after treatment showed decreased IR700 fluorescence in the tumors; since the therapeutic efficacy of NIR-PIT correlates with the degree of fluorescence loss of IR700 on APCs, the light dose at which fluorescence is completely lost in the tumor is interpreted as the saturation dose. In fluorescence monitoring studies of IR700 using the LIGHTVISION camera (Shimadzu, Kyoto, Japan), IR700 fluorescence declined rapidly after initial NIR exposure, reaching a plateau above a light dose of approximately 40–50 J/cm^2^ [22]. In both mouse models in this study, intratumoral fluorescence was still visible at 25 J/cm^2^ irradiation, but completely disappeared when the dose was raised to 50 J/cm^2^ or greater. In addition, increasing the light dose above 50 J/cm^2^ did not affect the TBR. Therefore, 50 J/cm^2^, the light dose at which the intratumor fluorescence disappeared completely, was considered to be the saturation dose in this experiment, and theoretically no improved therapeutic effect is expected by increasing the light dose above this saturation dose. The relationship between fluorescence imaging and therapeutic efficacy in an immunocompromised mouse model using the A431 tumor and Cetuximab has recently been reported [23]. In the article, the authors concluded that approximately 40 J/cm^2^ was the light dose with saturating NIR-PIT therapeutic effects. Although they observed tumor growth delay in the short term, the correlation between tumor-suppressing effects and the light dose was similar to our results. We also assessed the adverse effects of NIR-PIT. In both mouse models in this study, edema was observed after NIR-PIT and intensified with an increasing light dose. Moreover, the strong immune response and localized tumor response caused by NIR-PIT likely stressed the mice systemically. NIR-PIT causes minimal damage to normal cells due to the fact of its specific molecular targeting properties. However, in actual clinical practice, the formation of local edema around the tumor was observed early after treatment [24,25]. This edema is likely caused by ROS that is associated with acute inflammation caused by tumor or surrounding normal tissue damage [26,27], because such edema was suppressed by using reducing agents including ascorbic acid [14]. When NIR light is irradiated during NIR-PIT, ROS are generated by mostly unbound APCs under oxygen-rich conditions [28], especially in the blood flow that would cause edema. Energy diagrams of this photo-induced reaction of IR700 based on the analytical chemistry and the calculated reaction model are shown in previous studies [14,29]. ROS is not required for cell membrane damage induced by NIR-PIT but causes nonselective cytotoxicity and local edema following NIR-PIT. Therefore, excessive NIR light doses above the saturation dose may induce severe edema and should be avoided if possible.

There are several limitations to this study. First, only one cell line was tested as an immunocompetent mouse model. This was because mEERL-hEGFR, which expresses hEGFR, was the only model that can mimic clinical practice using humanized antibodies against human targets, yet normal mouse tissue does not express hEGFR. In order to elucidate the mechanism of edema formation in detail, it might be a good plan for us to inject APCs into nontumor-bearing mice. However, APC does not accumulate much less in normal skin and subcutaneous tissue than in tumor beds where enhanced permeability and retention (EPR) shows effects. Furthermore, the microenvironment in normal tissue and that in tumor beds are different. Therefore, we think it would be reasonable that mice bearing transplanted tumors are used as controls in order to evaluate the tumor microenvironment for simulating clinical conditions in cancer patients. Additionally, this study was not aimed at elucidating the mechanism of edema around NIR-PIT-treated tumors, which is a frequent side effect after NIR-PIT observed in most clinical patients. Second, we only tested different NIR light doses and kept the APC dose constant in this study. Treatment efficacy and adverse reactions may vary depending on the APC dose and treatment schedule as well. However, since this study focused on the light dose, we used a fixed APC dose that is commonly used in NIR-PIT. Third, tumors in humans are often larger than those found in mouse models. The appropriate light dose may vary depending on the tumor size; however, it is difficult to reproduce tumors of a similar size in rodents as in humans. In clinical practice, multiple light diffusers inserted into the tumors are used to compensate for the poor penetration of light into tissue. Fourth, we used different APCs for immunocompromised and immunocompetent mice. Panitumumab and Cetuximab are both antibodies against the overlapped epitope on the EGFR. Panitumumab, a fully humanized IgG2 antibody, has a longer half-life in serum than Cetuximab and has been our preferred choice for NIR-PIT evaluation in immunocompromised mouse models. On the other hand, Cetuximab, a chimeric IgG1 antibody, has higher antibody-dependent cellular cytotoxicity (ADCC) activity and is therefore more effective in immunocompetent mice. Cet-IR700 is used in clinical practice, and Cetuximab was used in this study in an immunocompetent mouse model to simulate clinical practice. Furthermore, it is the agent used in ongoing clinical trials of recurrent HNSCC, mimicking its use in humans. These antibodies were selected to maximize the therapeutic performance in each mouse model. Comparing the efficacy of NIR-PIT varying only the light dose in immunocompromised and immunocompetent mouse models may allow us to more accurate compare the effects of light. Finally, we did not evaluate different combinations of light intensity and light duration to produce the same light dose, but this variable may also influence treatment efficacy.

## 5. Conclusions

In an oncology clinic, a fixed dose of 50 J/cm^2^ is used as well as for surface exposure that frequently causes significant edema. Therefore, in order to determine an optimal light dose, this experiment was planned to use Cet-IR700 in immunocompetent mouse model for simulating clinical practice. In immunocompetent mouse models, irradiation of NIR light above the light dose for saturating therapeutic effects to cancer cells did not significantly improve tumor response but did induce more edema and systemic effects. The ability of NIR-PIT to activate mostly acquired host tumor immunity adds greater to its anti-tumor effect enabling lower doses of light to be used from a 0% cure rate in immunocompromised mice to 60–80% cure rate in immunocompetent mice with immune activation even after a single NIR-PIT [1,30,31]. The results suggested that 50 J/cm^2^ is a little too high for the surface illumination dose by considering both therapeutic effects and side effects. Therefore, in terms of optimizing the therapeutic efficacy and side effects, a relatively low NIR light dose, less than 50 J/cm^2^, is optimal for safely and efficiently treating tumors in NIR-PIT.

## Figures and Tables

**Figure 1 cancers-14-04042-f001:**
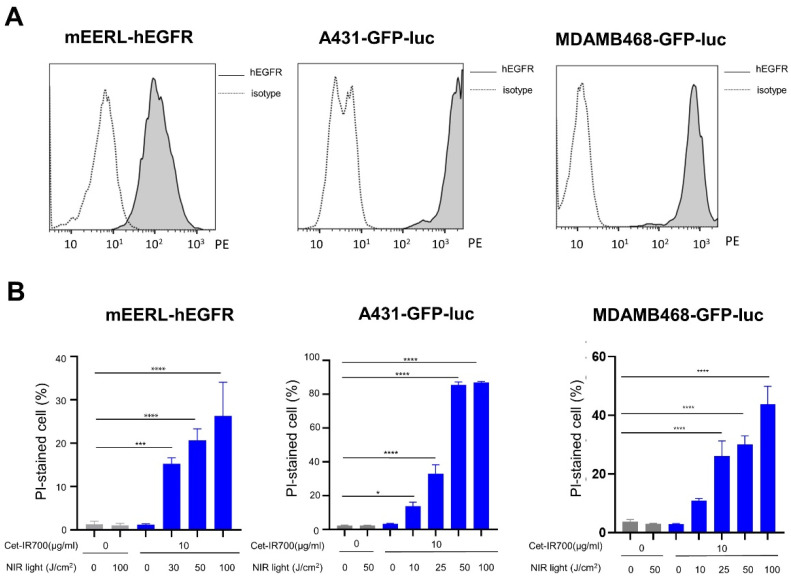
Efficacy of the NIR-PIT with Cet-IR700 in vitro: (**A**) In vitro hEGFR expression on mEERL-hEGFR, A431-GFP-luc, and MDAMB468-GFP-luc cells by flow cytometric analysis; (**B**) cell death induced by NIR-PIT with Cet-IR700 analyzed by flow cytometry using propidium iodide (PI) staining (*n* = 4; one-way ANOVA followed by Tukey’s test; *, *p* < 0.05; ***, *p* < 0.001; ****, *p* < 0.0001).

**Figure 2 cancers-14-04042-f002:**
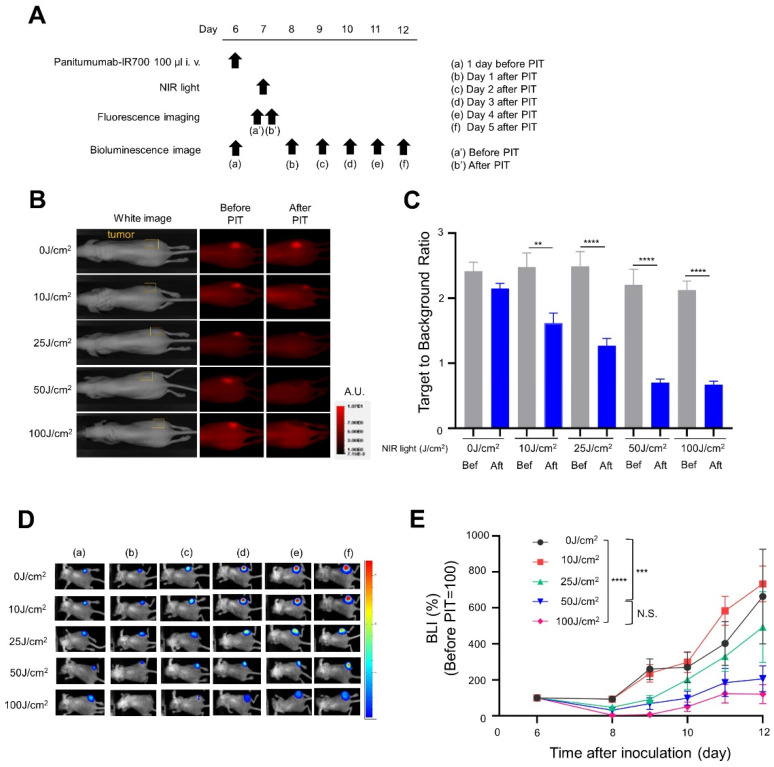
In vivo short-term efficacy of NIR-PIT with Pan-IR700 in A431-GFP-luc cells: (**A**) NIR-PIT regimen—bioluminescence and fluorescence images were obtained at each time point indicated; (**B**) fluorescence images for IR700 fluorescence—the yellow areas represent the location of the tumor (A.U., arbitrary unit); (**C**) target-to-background ratios (TBRs) of the IR700 fluorescence (*n* = 10; one-way ANOVA followed by Tukey’s test (**, *p* < 0.01 and ****, *p* < 0.0001; Bef: before; Aft: after); (**D**) Bioluminescence images were acquired in a series before and after NIR-PIT; (**E**) luciferase activity was quantified from BLI and shown as a relative percentage of the pretreatment signal intensity in each light dose treatment group (*n* = 10; one-way ANOVA followed by Tukey’s test; ***, *p* < 0.001 and ****, *p* < 0.0001; N.S.: not significant).

**Figure 3 cancers-14-04042-f003:**
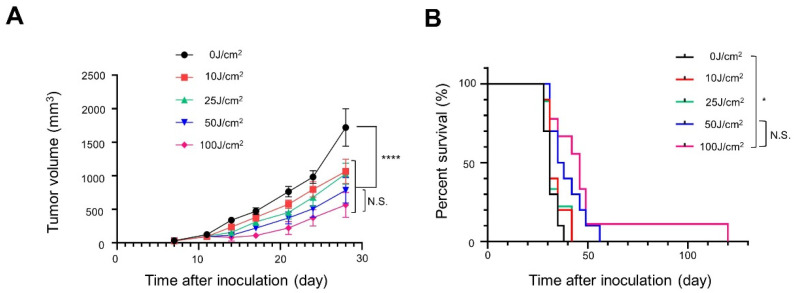
In vivo long-term efficacy of NIR-PIT with Pan-IR700 in A431-GFP-luc cells: (**A**) tumor volume curve (*n* = 10; one-way ANOVA followed by Tukey’s test; ****, *p* < 0.0001; 0 J/cm^2^ group vs. the other groups; N.S.: not significant; 50 J/cm^2^ vs. 100 J/cm^2^); (**B**) Kaplan–Meier survival curve (*n* = 10; log-rank test with Bonferroni correction; *, *p* < 0.05; N.S.: not significant).

**Figure 4 cancers-14-04042-f004:**
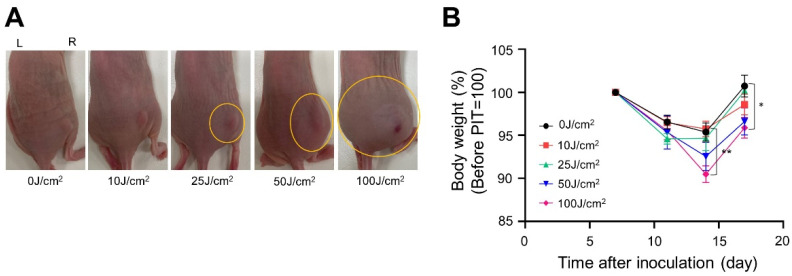
Side effects of the NIR-PIT with Pan-IR700 in A431-GFP-luc cells: (**A**) appearance on the treatment site 24 h after light exposure, where the yellow circles indicate the edema-affected area; (**B**) the percentage of weight change from immediately before the NIR light irradiation was measured (*n* = 10; one-way ANOVA followed by Tukey’s test; *, *p* < 0.05 and **, *p* < 0.01).

**Figure 5 cancers-14-04042-f005:**
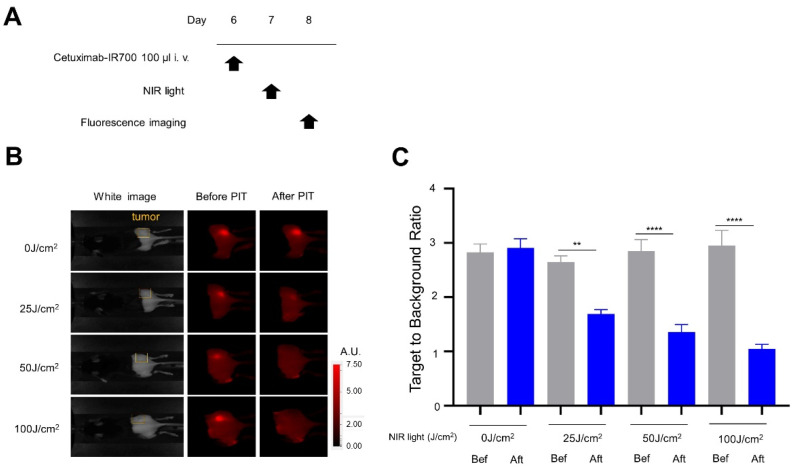
In vivo short-term efficacy of NIR-PIT with Cet-IR700 in mEERL-hEGFR tumors: (**A**) treatment schedule; (**B**) IR700 fluorescence images for IR700 fluorescence were obtained in each NIR light dose before and after NIR-PIT. The yellow areas represent the location of the tumor (A.U., arbitrary unit); (**C**) target-to-background ratio of IR700 fluorescence. (*n* = 10; one-way ANOVA followed by Tukey’s test; **, *p* < 0.01 and ****, *p* < 0.0001; Bef: before; Aft: after).

**Figure 6 cancers-14-04042-f006:**
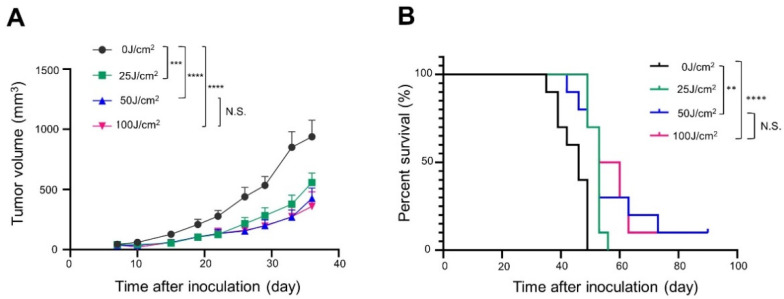
In vivo long-term efficacy of NIR-PIT with Cet-IR700 in mEERL-hEGFR tumors: (**A**) tumor volume curve (*n* = 10; one-way ANOVA followed by Tukey’s test; ***, *p* < 0.001 and ****, *p* < 0.0001; ns: not significant); (**B**) Kaplan–Meier survival curve (*n* = 10; log-rank test with Bonferroni correction; **, *p* < 0.01 and ****, *p* < 0.0001; N.S.: not significant).

**Figure 7 cancers-14-04042-f007:**
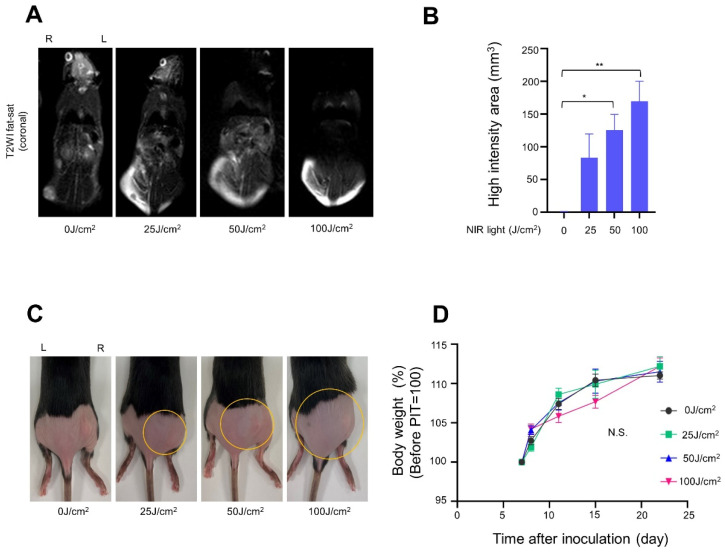
Side effect of NIR-PIT with Cet-IR700 in mEERL-hEGFR tumors: (**A**) edema was assessed by T2WI fat-sat MRI in the NIR-PIT-treated mice at 24 h; (**B**) the range of high signals was quantified using Image J (*n* = 3; one-way ANOVA followed by Tukey’s test; *, *p* < 0.05 and **, *p* < 0.01); (**C**) the appearance 24 h after light exposure, where the yellow circles indicate the edema-affected area; (**D**) the percentage of change from immediately before irradiation (*n* = 10; two-way ANOVA; N.S.: not significant).

## Data Availability

The data presented in this study is available on request from the corresponding author.

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
