# Peer review of "Optimal Light Dose for hEGFR-Targeted Near-Infrared Photoimmunotherapy"

_cancers, 2022, doi:10.3390/cancers14164042_

Round 1

Reviewer 1 Report

The authors evaluated anti-tumor effect and side effect after near-infrared photoimmunotherapy (NIR-PIT) using antibodies-conjugated photosensitizer (IR700) in immunocompetent and immunocompromised mice models bearing HGFR-high tumors. They concluded that development of severe local edema might be avoided by reducing illumination light intensity without decreasing anti-tumor effect of NIR-PIT since anti-tumor immune response could sufficiently be enhanced with smaller illumination light than conventional photodynamic therapy in the immunocompetent model and probably in clinical situations. Several concerns about study design and data interpretation should be resolved before publication.

 1. Cytokines other substances can be released not only from the tumor but also from surrounding tissues with residual IR700 after NIR irradiation. Therefore, the control group, i.e. non-tumor mice with administration of APC (IR700) and local NIR irradiation, would also be needed to evaluate efficacy of reducing illumination light power in avoiding local edema.

 2. Once antitumor immune response is really induced by NIR-PIT, it would affect on whole part of the tumor. However, at least a part of the antitumor effect after NIR-PIT would be due to local photothermal effect and radical oxygen like photodynamic treatment. Thus, the tissue damage in the deeper aspect of the tumor can be decreased when power of irradiation light is diminished in order to avoid local edema. Do the authors have cross sectional images of pathological examinations, suggesting difference of tissue damages between superficial and bottom regions of the tumor?

 3. Evaluation of local edema seems to be subjective (MRI would be enough but not used in the first series of experiments). Are there any standard methods available for the evaluation of edema grade (e.c. local temperature measured with a thermography)? Although body weight tended to decrease in the 100J/cm2 group, it would increase due to severe edema in the earlier period after NIR irradiation.

 4. Is there any possibilities to palliate edema after NIR-PIT by administration of vitamin C or other anti-inflammatory agents?

 5. Please demonstrate methods of NIR irradiation in more detail, in terms of name of the device, area and distance of irradiation, and duration of each session.

Author Response

  1. Cytokines other substances can be released not only from the tumor but also from surrounding tissues with residual IR700 after NIR irradiation. Therefore, the control group, i.e. non-tumor mice with administration of APC (IR700) and local NIR irradiation, would also be needed to evaluate efficacy of reducing illumination light power in avoiding local edema.

>As the reviewer pointed out, in order to elucidate the mechanism of edema formation in detail, it might be a good for us to inject APCs into non-tumor bearing mice, although APC does not accumulate much less in normal skin than in tumor beds where show EPR effects. However, since this study is not aimed for elucidating the mechanism of edema around NIR-PIT treated tumors that is a frequent side effect of NIR-PIT. As the reviewer pointed out, TME in normal cells and TME in the presence of tumor cells may be different. Therefore, we think it would be reasonable that tumor transplanted mice were used as controls in order to evaluate them under more clinical conditions.

  1. Once antitumor immune response is really induced by NIR-PIT, it would affect on whole part of the tumor. However, at least a part of the antitumor effect after NIR-PIT would be due to local photothermal effect and radical oxygen like photodynamic treatment. Thus, the tissue damage in the deeper aspect of the tumor can be decreased when power of irradiation light is diminished in order to avoid local edema. Do the authors have cross sectional images of pathological examinations, suggesting difference of tissue damages between superficial and bottom regions of the tumor?

>Sufficient therapeutic dose of NIR light for NIR-PIT typically penetrates up to 2 cm. As the reviewer pointed out, NIR also attenuates with depth. Therefore, tissue injury is likely to be less severe in deeper areas. A paper on this NIR light attenuation was written by Shuhei Okuyama in Oncotarget 2018 Jan 27;9(13):11159-11169. Moreover, this paper shows that the higher the dose, the deeper the penetration into the tissue. Furthermore, the enhancing therapeutic effects to the deep part of tumors with increasing light dose is also demonstrated by the histology that was described in the supplemental figure of Furusawara's paper. (Furusawa et al. Oncoimmunology. 2022 Jan 4;11(1):2019922.)

  1. Evaluation of local edema seems to be subjective (MRI would be enough but not used in the first series of experiments). Are there any standard methods available for the evaluation of edema grade (e.c. local temperature measured with a thermography)? Although body weight tended to decrease in the 100J/cm2group, it would increase due to severe edema in the earlier period after NIR irradiation.

>It is difficult to assess the total amount of edema by temperature because surface temperature depends on causes of edema including inflammation or vascular leakage. MRI appears to be quick and accurate way to quantitatively assess edema especially in the human body. In the present project, an immunocompetent mouse model, which is a better model for simulate a clinical setting, was used for the detailed evaluation. Regarding the body weight change, we did not measure the body weight on the day after irradiation in the immunocompetent model. As the reviewer indicated, it would be possible that body weight was temporarily increased on the following day.

  1. Is there any possibilities to palliate edema after NIR-PIT by administration of vitamin C or other anti-inflammatory agents?

>Please read the article by Kato.et.al. (Kato.et.al in ACS Pharmacol Transl Sci. 2021 Sep 17;4(5):1689-1701.). Theoretically, any reducing agent can reduce edema, but various current studies have shown that ascorbic acid is the best reducing agent for suppressing edema but unaffecting the NIR-PIT therapeutic effects.

  1. Please demonstrate methods of NIR irradiation in more detail, in terms of name of the device, area and distance of irradiation, and duration of each session.

>We have added this point to Material & Methods. (Line 167-170)

Reviewer 2 Report

In this manuscript by Hideyuki Furumoto, et al., entitled "Optimal Light Dose For hEGFR-targeted Near-infrared Photoimmunotherapy" demonstrated tumor growth suppression by anti-EGFR antibodies with IR700Dye in an immunodeficient and immunocompetent mouse model. The authors found that the amount of light irradiation can be reduced, leading to a reduction in side effects in an immunocompetent mouse model. Although the study progress was adequate, the manuscript is needed to be extensively improved.

Major points:

1. In Fig.1, the authors use three cell lines (mEERL-hEGFR, A431-GFP-luc, MDAMB468-GFP-luc). The authors should indicate the EGFR expression levels of these cell lines and discuss the relationship between the amount of EGFR expression and the cell-killing effect.

2. This paper uses two anti-EGFR antibodies, cetuximab and panitumumab, why was in vitro testing done only with cetuximab? It is necessary to present panitumumab data to show that the fabricated Pani-IR700 works properly.

3. The authors discuss edema after NIR-PIT. In the first place, please provide any data or literature on whether edema occurs when the unconjugated naked antibody treated group or untreated group is exposed to high doses of NIR light. In my opinion, 100 J/cm2 would be the intensity at which burns could occur on the skin of the mouse. If NIR light alone has no side effects, it should be shown.

4. In Fig.4B and 7D, the authors showed body weight change of mice by the actual amount. In particular, the body weight of mice implanted with tumors should be analyzed relatively, since there are individual differences.

5. I found a paper that already reported examining laser intensity and anti-tumor effects against A431 cell lines with cetuximab-IR700 (Takashima et al., Pharmaceuticals 2022, 15(2), 223; https://doi.org/10.3390/ph15020223). The authors need to discuss it.

6. The authors have already reported a paper on anti-EGFR antibody selection and light irradiation regimens (Okada et al., Cancer immunology, immunotherapy: CII 2022, doi:10.1007/s00262-021-03124-x. ref.5). I understand that this reviewing manuscript is a different approach from the previous reports. However, since the authors are using the same Cetuximab-IR700 or Panitumumab-IR700, and mEERL-hEGFR cell line, please describe in the Discussion section what the authors think is the best regimen and drug at the moment, after all, based on previous findings and new findings in this manuscript.

Minor points:

1. line:201 and Figure1, MDA-MB-468 is written as MDAMD468. Please fix it.

2. In Fig.4B, there is no legend for the graph, please indicate.

3. In Fig.7B, “High indensity area” is correct? intensity?

Author Response

Major points:

  1. In Fig.1, the authors use three cell lines (mEERL-hEGFR, A431-GFP-luc, MDAMB468-GFP-luc). The authors should indicate the EGFR expression levels of these cell lines and discuss the relationship between the amount of EGFR expression and the cell-killing effect.

>It has been proven from many previous NIR-PIT publications that the expression level is basically proportional to the cell-killing effect. we have added a histogram showing the expression levels of hEGFR in three cell lines in Fig. 1A. The results show that the higher the expression level, the higher the cell-killing effect at least in vitro.

  1. This paper uses two anti-EGFR antibodies, cetuximab and panitumumab, why was in vitro testing done only with cetuximab? It is necessary to present panitumumab data to show that the fabricated Pani-IR700 works properly.

 >A comparison of panitumumab and cetuximab is discussed in the following paper written by Okada et al. in Cancer Immunology Immunotherapy volume 71, pages1877–1887 (2022)) The data showed that Pan-IR700 has a little better therapeutic potential than cet-IR700. Currently, NIR-PIT is approved for cancer therapy under health insurance in Japan. A phase III trial is also underway worldwide. The currently used APC are made with cet-IR700. We are studying it with Cet-IR700 because it is the one used in clinical practice and we wanted to simulate it.

  1. The authors discuss edema after NIR-PIT. In the first place, please provide any data or literature on whether edema occurs when the unconjugated naked antibody treated group or untreated group is exposed to high doses of NIR light. In my opinion, 100 J/cm2 would be the intensity at which burns could occur on the skin of the mouse. If NIR light alone has no side effects, it should be shown.

>First, edema in NIR-PIT is not caused by burns but by increased vascular permeability. This is described in the paper by Kato et al. Therefore, in this experiment, the evaluation of edema was compared using the light dose as a variable after the clinical dose of APC administration. (Kato.et.al in ACS Pharmacol Transl Sci. 2021 Sep 17;4(5):1689-1701.)

 In the previous stidies below, NIR-PIT can be safely performed with near-infrared light exposure up to a power density of 600 mW/cm2. The thermal displacement of 100 J/cm NIR irradiated at 330 mW/cm2 was negligible and no thermal damage occurred, as described in detail in the following paper on thermogenesis. This experiment was performed at a lower power density of 150 mW/cm2 that is the dose used in clinic.

(Shuhei Okuyama et al in Oncotarget 2017 Aug 11;8(68):113194-113201,

Shuhei Okuyama in Oncotarget 2018 Jan 27;9(13):11159-11169)

  1. In Fig.4B and 7D, the authors showed body weight change of mice by the actual amount. In particular, the body weight of mice implanted with tumors should be analyzed relatively, since there are individual differences.

 >As the reviewer pointed out, we agreed to compare the data in the individual mice. The data has been edited and these figures were replaced.

  1. I found a paper that already reported examining laser intensity and anti-tumor effects against A431 cell lines with cetuximab-IR700 (Takashima et al., Pharmaceuticals2022, 15(2), 223; https://doi.org/10.3390/ph15020223). The authors need to discuss it.

>The first real-time light vision paper written by Okuyama et al. and the paper you suggested show that the IR700 fluorescence decay plateaus at around 40 J/cm2. (Shuhei Okuyama et al in Cancer Diagn Progn 2021 May 3;1(2):29-34) The anti-tumor effects reported in the suggested paper shows shorter follow-up than this work, yet both results are almost identical. However, my paper concludes that the therapeutic effect reaches equilibrium at around 40 J/cm2 light dose, while side effects increase in a light dose-dependent manner, and therefore, high light doses should not be used. Therefore, the conclusion are altered with light dose-dependent side effects.

  1. The authors have already reported a paper on anti-EGFR antibody selection and light irradiation regimens (Okada et al., Cancer immunology, immunotherapy: CII 2022, doi:10.1007/s00262-021-03124-x. ref.5). I understand that this reviewing manuscript is a different approach from the previous reports. However, since the authors are using the same Cetuximab-IR700 or Panitumumab-IR700, and mEERL-hEGFR cell line, please describe in the Discussion section what the authors think is the best regimen and drug at the moment, after all, based on previous findings and new findings in this manuscript.

>Okada et al. reported that two 30 J/cm2 irradiations with panitumumab were effective. Actually, panitumumab is greater effect than cetuximab. In oncology clinic, 50 J/cm2 is used and significant edema is observed. Therefore, this experiment was evaluated with cetuximab-IR700 simulating clinical practice. The results suggested that 50 J/cm2 is too high for a surface irradiation dose considering both therapeutic effects and side effects. From the results of the experiments focusing on irradiation dose, NIR-PIT is sufficiently effective even at a little lower light doses than the current clinical dose ,50 J/cm2, and side effects can be minimized.

Minor points:

  1. line:201 and Figure1, MDA-MB-468 is written as MDAMD468. Please fix it.

>Thank you for pointing this out. It has been corrected.

  1. In Fig.4B, there is no legend for the graph, please indicate.

 >Thank you for pointing this out. It has been added

  1. In Fig.7B, “High indensity area” is correct? intensity?

>Thank you for pointing this out. It has been corrected.

Round 2

Reviewer 1 Report

The manuscript has been revised accordingly. I would recommend the authors to add responses to Q1 to discussion section in the manuscript.

Author Response

The manuscript has been revised accordingly. I would recommend the authors to add responses to Q1 to discussion section in the manuscript.

  • We have added the Q1 response comment as a limitation in the Discussion section. (Line 384-392)

Reviewer 2 Report

Thank you for sending the revised manuscript. Some points have been improved, but many corrections still need to be made before publication.

Major points:

  1. In Fig.1, the authors use three cell lines (mEERL-hEGFR, A431-GFP-luc, MDAMB468-GFP-luc). The authors should indicate the EGFR expression levels of these cell lines and discuss the relationship between the amount of EGFR expression and the cell-killing effect.

>It has been proven from many previous NIR-PIT publications that the expression level is basically proportional to the cell-killing effect. we have added a histogram showing the expression levels of hEGFR in three cell lines in Fig. 1A. The results show that the higher the expression level, the higher the cell-killing effect at least in vitro.

I think the revision makes the diagram easier for the reader to understand.

Please fill the values of the horizontal axis in Fig. 1A.

  1. This paper uses two anti-EGFR antibodies, cetuximab and panitumumab, why was in vitro testing done only with cetuximab? It is necessary to present panitumumab data to show that the fabricated Pani-IR700 works properly.

>A comparison of panitumumab and cetuximab is discussed in the following paper written by Okada et al. in Cancer Immunology Immunotherapy volume 71, pages1877–1887 (2022)) The data showed that Pan-IR700 has a little better therapeutic potential than cet-IR700. Currently, NIR-PIT is approved for cancer therapy under health insurance in Japan. A phase III trial is also underway worldwide. The currently used APC are made with cet-IR700. We are studying it with Cet-IR700 because it is the one used in clinical practice and we wanted to simulate it.

“We are studying it with Cet-IR700 because it is the one used in clinical practice and we wanted to simulate it.” If so, why are the authors use Pan-IR700 in Figure 2-4? If you only focus on cetuximab, shouldn't you get the artificial IgG2 type cetuximab (e.g. https://www.invivogen.com/anti-hegfr-higg2) and experiment with it? I could not understand your explanation. What is the purpose of the in vitro experiment in Figure 1? I understood that you are doing this to see if the antibody-IR700 you created worked properly for the three cell lines used in this paper. Therefore, I pointed out that the same experiment should be done with Pan-IR700.

  1. The authors discuss edema after NIR-PIT. In the first place, please provide any data or literature on whether edema occurs when the unconjugated naked antibody treated group or untreated group is exposed to high doses of NIR light. In my opinion, 100 J/cm2 would be the intensity at which burns could occur on the skin of the mouse. If NIR light alone has no side effects, it should be shown.

>First, edema in NIR-PIT is not caused by burns but by increased vascular permeability. This is described in the paper by Kato et al. Therefore, in this experiment, the evaluation of edema was compared using the light dose as a variable after the clinical dose of APC administration. (Kato.et.al in ACS Pharmacol Transl Sci. 2021 Sep 17;4(5):1689-1701.)
In the previous stidies below, NIR-PIT can be safely performed with near-infrared light exposure up to a power density of 600 mW/cm2. The thermal displacement of 100 J/cm NIR irradiated at 330 mW/cm2 was negligible and no thermal damage occurred, as described in detail in the following paper on thermogenesis. This experiment was performed at a lower power density of 150 mW/cm2 that is the dose used in clinic. (Shuhei Okuyama et al in Oncotarget 2017 Aug 11;8(68):113194-113201,Shuhei Okuyama in Oncotarget 2018 Jan 27;9(13):11159-11169)

I carefully read the paper presented by the authors (Takuya Kato et al). There is "ROS increases vascular permeability" in the discussion, but no mention about "edema in NIR-PIT is not caused by burns". This paper compares the presence or absence of L-NaAA, but does not examine whether edema occurs in the absence of IR700Dye in the first place.

I read two Oncotarget papers presented by the authors. In 2017 paper, no increase in skin temperature has been observed when irradiated at a total energy of 60 J/cm2 and a density of 150 mW/cm2. However, in Fig. 4 of this manuscript, edema is seen at a total energy of 25 J/cm2 and a density of 150 mW/cm2. Why is this? My point is that there is no proper “negative” control to see side effects. At least, you should add the data with unconjugated naked antibody treated group or untreated group is exposed to 100J/cm2. If you want to mention side effects, you need to show that there are no side effects from near infrared alone in this condition.

  1. In Fig.4B and 7D, the authors showed body weight change of mice by the actual amount. In particular, the body weight of mice implanted with tumors should be analyzed relatively, since there are individual differences.

>As the reviewer pointed out, we agreed to compare the data in the individual mice. The data has been edited and these figures were replaced.

OK, but the vertical axis should be displayed as either % or 1.

  1. I found a paper that already reported examining laser intensity and anti-tumor effects against A431 cell lines with cetuximab-IR700 (Takashima et al., Pharmaceuticals2022, 15(2), 223; https://doi.org/10.3390/ph15020223). The authors need to discuss it.

>The first real-time light vision paper written by Okuyama et al. and the paper you suggested show that the IR700 fluorescence decay plateaus at around 40 J/cm2. (Shuhei Okuyama et al in Cancer Diagn Progn 2021 May 3;1(2):29-34) The anti-tumor effects reported in the suggested paper shows shorter follow-up than this work, yet both results are almost identical. However, my paper concludes that the therapeutic effect reaches equilibrium at around 40 J/cm2 light dose, while side effects increase in a light dose-dependent manner, and therefore, high light doses should not be used. Therefore, the conclusion are altered with light dose-dependent side effects.

Comparison of the two light vision papers is not necessary within this manuscript. However, the paper I found uses the same antibodies and the same cells, and also analyzes the histopathology, which is important because it has a different angle than this manuscript. Since the authors focused on determining the optimal light intensity to balance treatment efficacy and side effects, you should cite this article appropriately and describe it in the Discussion.

  1. The authors have already reported a paper on anti-EGFR antibody selection and light irradiation regimens (Okada et al., Cancer immunology, immunotherapy: CII 2022, doi:10.1007/s00262-021-03124-x. ref.5). I understand that this reviewing manuscript is a different approach from the previous reports. However, since the authors are using the same Cetuximab-IR700 or Panitumumab-IR700, and mEERL-hEGFR cell line, please describe in the Discussion section what the authors think is the best regimen and drug at the moment, after all, based on previous findings and new findings in this manuscript.

>Okada et al. reported that two 30 J/cm2 irradiations with panitumumab were effective. Actually, panitumumab is greater effect than cetuximab. In oncology clinic, 50 J/cm2 is used and significant edema is observed. Therefore, this experiment was evaluated with cetuximab-IR700 simulating clinical practice. The results suggested that 50 J/cm2 is too high for a surface irradiation dose considering both therapeutic effects and side effects. From the results of the experiments focusing on irradiation dose, NIR-PIT is sufficiently effective even at a little lower light doses than the current clinical dose ,50 J/cm2, and side effects can be minimized.

Please describe this argument in the Discussion section in the manuscript. It is important for the reader to compare a lot of previous studies from the authors group with the "new" findings in this paper.

Minor points:

1, new line:146, anti-human EGFR Ab is written as anti-human EFGR Ab. Please fix it.

2, In section 2.11, it says "one-way analysis of variance (ANOVA) followed by the Bonferroni correction for multiple comparisons was used", but in the figure legends, " One-way ANOVA followed by Tukey's test ". Please confirm.

3, Figure resolution is low, please correct.

Author Response

Thank you for sending the revised manuscript. Some points have been improved, but many corrections still need to be made before publication.

>Thank you so much for reviewing our article. We have responded as much as we could do as shown below.

Major points:

  1. In Fig.1, the authors use three cell lines (mEERL-hEGFR, A431-GFP-luc, MDAMB468-GFP-luc). The authors should indicate the EGFR expression levels of these cell lines and discuss the relationship between the amount of EGFR expression and the cell-killing effect.

>It has been proven from many previous NIR-PIT publications that the expression level is basically proportional to the cell-killing effect. we have added a histogram showing the expression levels of hEGFR in three cell lines in Fig. 1A. The results show that the higher the expression level, the higher the cell-killing effect at least in vitro.

I think the revision makes the diagram easier for the reader to understand.

Please fill the values of the horizontal axis in Fig. 1A.

>We have filled the value of x-axis in Fig 1A as shown.

  1. This paper uses two anti-EGFR antibodies, cetuximab and panitumumab, why was in vitro testing done only with cetuximab? It is necessary to present panitumumab data to show that the fabricated Pani-IR700 works properly.

>A comparison of panitumumab and cetuximab is discussed in the following paper written by Okada et al. in Cancer Immunology Immunotherapy volume 71, pages1877–1887 (2022)) The data showed that Pan-IR700 has a little better therapeutic potential than cet-IR700. Currently, NIR-PIT is approved for cancer therapy under health insurance in Japan. A phase III trial is also underway worldwide. The currently used APC are made with cet-IR700. We are studying it with Cet-IR700 because it is the one used in clinical practice and we wanted to simulate it.

“We are studying it with Cet-IR700 because it is the one used in clinical practice and we wanted to simulate it.” If so, why are the authors use Pan-IR700 in Figure 2-4? If you only focus on cetuximab, shouldn't you get the artificial IgG2 type cetuximab (e.g. https://www.invivogen.com/anti-hegfr-higg2) and experiment with it? I could not understand your explanation. What is the purpose of the in vitro experiment in Figure 1? I understood that you are doing this to see if the antibody-IR700 you created worked properly for the three cell lines used in this paper. Therefore, I pointed out that the same experiment should be done with Pan-IR700.

>As described in our previous works, these two antibodies recognized an overlapped epitope on the EGFR and identically worked on NIR-PIT to EGFR-expressing cells in vitro. The purpose of Fig. 1 is to show that light dose-dependent NIR-PIT effects to these EGFR-expressing cells in vitro in order to validate EGFR-targeting NIR-PIT worked on these cells. In the Discussion line 378-394, we have edited to clarify advantages and limitations of panitumumab and cetuximab that was the reason why we selected the Ab for each experiment.                                                         

  1. The authors discuss edema after NIR-PIT. In the first place, please provide any data or literature on whether edema occurs when the unconjugated naked antibody treated group or untreated group is exposed to high doses of NIR light. In my opinion, 100 J/cm2 would be the intensity at which burns could occur on the skin of the mouse. If NIR light alone has no side effects, it should be shown.

>First, edema in NIR-PIT is not caused by burns but by increased vascular permeability. This is described in the paper by Kato et al. Therefore, in this experiment, the evaluation of edema was compared using the light dose as a variable after the clinical dose of APC administration. (Kato.et.al in ACS Pharmacol Transl Sci. 2021 Sep 17;4(5):1689-1701.)
In the previous stidies below, NIR-PIT can be safely performed with near-infrared light exposure up to a power density of 600 mW/cm2. The thermal displacement of 100 J/cm NIR irradiated at 330 mW/cm2 was negligible and no thermal damage occurred, as described in detail in the following paper on thermogenesis. This experiment was performed at a lower power density of 150 mW/cm2 that is the dose used in clinic.
 (Shuhei Okuyama et al in Oncotarget 2017 Aug 11;8(68):113194-113201,Shuhei Okuyama in Oncotarget 2018 Jan 27;9(13):11159-11169)

I carefully read the paper presented by the authors (Takuya Kato et al). There is "ROS increases vascular permeability" in the discussion, but no mention about "edema in NIR-PIT is not caused by burns". This paper compares the presence or absence of L-NaAA, but does not examine whether edema occurs in the absence of IR700Dye in the first place.

I read two Oncotarget papers presented by the authors. In 2017 paper, no increase in skin temperature has been observed when irradiated at a total energy of 60 J/cm2 and a density of 150 mW/cm2. However, in Fig. 4 of this manuscript, edema is seen at a total energy of 25 J/cm2 and a density of 150 mW/cm2. Why is this? My point is that there is no proper “negative” control to see side effects. At least, you should add the data with unconjugated naked antibody treated group or untreated group is exposed to 100J/cm2. If you want to mention side effects, you need to show that there are no side effects from near infrared alone in this condition.

> We agree with the reviewer that the cause of edema is an interesting topic. Since the edema is one of severe adverse effects after NIR-PIT in clinic, we should care about the severity of edema when tumors are treated with NIR-PIT. In this paper, we demonstrated that severity of edema is well correlated to light dose. Additionally, our previous work demonstrated that edema mostly suppressed with injection of ascorbic acid as a reducing agent. Along the line, although the suggested experiment without tumor by the reviewer would be interesting, since this paper is focusing on optimal light exposure dose of NIR-PIT based on both therapeutic effects and adverse effects, the non-tumor experiment is out of our focus of this paper. We are still working on the project to minimize the edema using reducing agents. We will try such experiment in our future projects to more precisely investigate the cause of edema.

I read two Oncotarget papers presented by the authors. In 2017 paper, no increase in skin temperature has been observed when irradiated at a total energy of 60 J/cm2 and a density of 150 mW/cm2. However, in Fig. 4 of this manuscript, edema is seen at a total energy of 25 J/cm2 and a density of 150 mW/cm2. Why is this? 

>Our previous results suggested that edema by NIR-PIT was not caused by the elevated local skin temperature but caused by ROS production as described in Kato’s paper (Kato.et.al in ACS Pharmacol Transl Sci. 2021 Sep 17;4(5):1689-1701.).

  1. In Fig.4B and 7D, the authors showed body weight change of mice by the actual amount. In particular, the body weight of mice implanted with tumors should be analyzed relatively, since there are individual differences.

>As the reviewer pointed out, we agreed to compare the data in the individual mice. The data has been edited and these figures were replaced.

OK, but the vertical axis should be displayed as either % or 1.

> Thank you for pointing this out. We have revised the Fig 4B and 7D that the individual mouse weight before PIT was set as 100, and then showed the percentage of body weight after the NIR-PIT; Body weight (%) (before PIT=100)

  1. I found a paper that already reported examining laser intensity and anti-tumor effects against A431 cell lines with cetuximab-IR700 (Takashima et al., Pharmaceuticals2022, 15(2), 223; https://doi.org/10.3390/ph15020223). The authors need to discuss it.

>The first real-time light vision paper written by Okuyama et al. and the paper you suggested show that the IR700 fluorescence decay plateaus at around 40 J/cm2. (Shuhei Okuyama et al in Cancer Diagn Progn 2021 May 3;1(2):29-34) The anti-tumor effects reported in the suggested paper shows shorter follow-up than this work, yet both results are almost identical. However, my paper concludes that the therapeutic effect reaches equilibrium at around 40 J/cm2 light dose, while side effects increase in a light dose-dependent manner, and therefore, high light doses should not be used. Therefore, the conclusion are altered with light dose-dependent side effects.

Comparison of the two light vision papers is not necessary within this manuscript. However, the paper I found uses the same antibodies and the same cells, and also analyzes the histopathology, which is important because it has a different angle than this manuscript. Since the authors focused on determining the optimal light intensity to balance treatment efficacy and side effects, you should cite this article appropriately and describe it in the Discussion.

 >Thank you for introducing this new important paper to us. We have added this issue to the discussion by citing this suggested references as new ref #23.

  1. The authors have already reported a paper on anti-EGFR antibody selection and light irradiation regimens (Okada et al., Cancer immunology, immunotherapy: CII 2022, doi:10.1007/s00262-021-03124-x. ref.5). I understand that this reviewing manuscript is a different approach from the previous reports. However, since the authors are using the same Cetuximab-IR700 or Panitumumab-IR700, and mEERL-hEGFR cell line, please describe in the Discussion section what the authors think is the best regimen and drug at the moment, after all, based on previous findings and new findings in this manuscript.

>Okada et al. reported that two 30 J/cm2 irradiations with panitumumab were effective. Actually, panitumumab is greater effect than cetuximab. In oncology clinic, 50 J/cm2 is used and significant edema is observed. Therefore, this experiment was evaluated with cetuximab-IR700 simulating clinical practice. The results suggested that 50 J/cm2 is too high for a surface irradiation dose considering both therapeutic effects and side effects. From the results of the experiments focusing on irradiation dose, NIR-PIT is sufficiently effective even at a little lower light doses than the current clinical dose ,50 J/cm2, and side effects can be minimized.

Please describe this argument in the Discussion section in the manuscript. It is important for the reader to compare a lot of previous studies from the authors group with the "new" findings in this paper.

> We have added the following to the conclusion section

Minor points:

1, new line:146, anti-human EGFR Ab is written as anti-human EFGR Ab. Please fix it.

>Thank you. Corrected

2, In section 2.11, it says "one-way analysis of variance (ANOVA) followed by the Bonferroni correction for multiple comparisons was used", but in the figure legends, " One-way ANOVA followed by Tukey's test ". Please confirm.

>Thank you. Corrected

3, Figure resolution is low, please correct.

>We have revised the figures to increase the resolution.

Round 3

Reviewer 2 Report

Thank you for sending the second round revised manuscript. 

Most points have been improved. However, I think a few more corrections still need to be made before publication.

Major point 2

>As described in our previous works, these two antibodies recognized an overlapped epitope on the EGFR and identically worked on NIR-PIT to EGFR-expressing cells in vitro. The purpose of Fig. 1 is to show that light dose-dependent NIR-PIT effects to these EGFR-expressing cells in vitro in order to validate EGFR-targeting NIR-PIT worked on these cells. In the Discussion line 378-394, we have edited to clarify advantages and limitations of panitumumab and cetuximab that was the reason why we selected the Ab for each experiment.

As the authors are well known, cetuximab and panitumumab have the same mechanism of action but differ in epitopes and binding activity. Furthermore, only Cet-IR700 is approved for NIR-PIT. I understand that Cet-IR700 appears in 3.1 as to validate EGFR-targeting NIR-PIT, but it is strange to see Pani-IR700 with another antibody in 3.2 without any explanation. I think that the purpose of the experiment (to validate EGFR-targeting NIR-PIT) should be added at the beginning of 3.1 and the reason for using Pan-IR700 instead of Cet-IR700 should be stated at the beginning of 3.2.

Major point 3

> We agree with the reviewer that the cause of edema is an interesting topic. Since the edema is one of severe adverse effects after NIR-PIT in clinic, we should care about the severity of edema when tumors are treated with NIR-PIT. In this paper, we demonstrated that severity of edema is well correlated to light dose. Additionally, our previous work demonstrated that edema mostly suppressed with injection of ascorbic acid as a reducing agent. Along the line, although the suggested experiment without tumor by the reviewer would be interesting, since this paper is focusing on optimal light exposure dose of NIR-PIT based on both therapeutic effects and adverse effects, the non-tumor experiment is out of our focus of this paper. We are still working on the project to minimize the edema using reducing agents. We will try such experiment in our future projects to more precisely investigate the cause of edema.

I understand that you have demonstrated that the severity of edema correlates well with light dose. And also, it is fascinating and very important that small molecule compounds can reduce ROS, and I look forward to seeing a paper on this in future work. However, I am not demanding an experiment “in the absence of tumors”, but rather that it should be properly shown as a negative control that high dose light irradiation alone does not cause edema “in the absence of drug” (unconjugated naked antibody treated group or untreated group). In this paper, you must first show that edema does not occur in the absence of IR700 Dye and only with high dose laser irradiation (100 J/cm2), or you cannot discuss whether it is an IR700-induced or laser-induced side effect. Once again, it should be shown whether light irradiation alone cannot cause edema in the absence of drugs, regardless of the presence or absence of tumor. This point is critical.

Author Response

Thank you for sending the second round revised manuscript. 

Most points have been improved. However, I think a few more corrections still need to be made before publication.

Major point 2

>As described in our previous works, these two antibodies recognized an overlapped epitope on the EGFR and identically worked on NIR-PIT to EGFR-expressing cells in vitro. The purpose of Fig. 1 is to show that light dose-dependent NIR-PIT effects to these EGFR-expressing cells in vitro in order to validate EGFR-targeting NIR-PIT worked on these cells. In the Discussion line 378-394, we have edited to clarify advantages and limitations of panitumumab and cetuximab that was the reason why we selected the Ab for each experiment.

As the authors are well known, cetuximab and panitumumab have the same mechanism of action but differ in epitopes and binding activity. Furthermore, only Cet-IR700 is approved for NIR-PIT. I understand that Cet-IR700 appears in 3.1 as to validate EGFR-targeting NIR-PIT, but it is strange to see Pani-IR700 with another antibody in 3.2 without any explanation. I think that the purpose of the experiment (to validate EGFR-targeting NIR-PIT) should be added at the beginning of 3.1 and the reason for using Pan-IR700 instead of Cet-IR700 should be stated at the beginning of 3.2.

  • We have explained this point at the beginning of both sections 3.2 (Line 232-235) and 3.3 (Line 282).

Major point 3

> We agree with the reviewer that the cause of edema is an interesting topic. Since the edema is one of severe adverse effects after NIR-PIT in clinic, we should care about the severity of edema when tumors are treated with NIR-PIT. In this paper, we demonstrated that severity of edema is well correlated to light dose. Additionally, our previous work demonstrated that edema mostly suppressed with injection of ascorbic acid as a reducing agent. Along the line, although the suggested experiment without tumor by the reviewer would be interesting, since this paper is focusing on optimal light exposure dose of NIR-PIT based on both therapeutic effects and adverse effects, the non-tumor experiment is out of our focus of this paper. We are still working on the project to minimize the edema using reducing agents. We will try such experiment in our future projects to more precisely investigate the cause of edema.

I understand that you have demonstrated that the severity of edema correlates well with light dose. And also, it is fascinating and very important that small molecule compounds can reduce ROS, and I look forward to seeing a paper on this in future work. However, I am not demanding an experiment “in the absence of tumors”, but rather that it should be properly shown as a negative control that high dose light irradiation alone does not cause edema “in the absence of drug” (unconjugated naked antibody treated group or untreated group). In this paper, you must first show that edema does not occur in the absence of IR700 Dye and only with high dose laser irradiation (100 J/cm2), or you cannot discuss whether it is an IR700-induced or laser-induced side effect. Once again, it should be shown whether light irradiation alone cannot cause edema in the absence of drugs, regardless of the presence or absence of tumor. This point is critical.

  • We performed suggested experiments using A431 tumor bearing mouse model with 0, 10 and 100 mg of APC and similar light doses with different fluency as shown in our previously published work (Fig. 3 in Okuyama S, Oncotarget 2017; 8(68): 113194-113201.). Although we did not describe the edema in detail, we observed no edema in all mice of APC 0 mg group. Therefore, we never observed edema up to 600 mW/100 J/cm2 of light only (no APC) that did not induce any thermal damage in the skin. Additionally, in order to reflect the reviewer’s suggestion, we have added several sentences for explaining the reason why we did not use non-tumor bearing mice at the limitation in the Discussion section. (Line 384-392)